# MIdASv0.2.1—MultI-scale bias AdjuStment

Peter Berg[1,*], Thomas Bosshard[1,*], Wei Yang[1,*], and Klaus Zimmermann[1,*]

[1]Swedish Meteorological and Hydrological Institute, Folkborgsvägen 17, 601 76 Norrköping, Sweden
[*]These authors contributed equally to this work.

**Correspondence:** Peter Berg (peter.berg@smhi.se)

**Abstract.** Bias adjustment is the practice of statistically transforming climate model data in order to reduce systematic deviations from a reference data set, typically some sort of observations. There are numerous proposed methodologies to perform the adjustments – ranging from simple scaling approaches to advanced multi-variate distribution based mapping. In practice, the actual bias adjustment method is a small step in the application, and most of the processing handles reading, writing and linking different data sets. These practical processing steps become especially heavy with increasing model domain size and resolution in both time and space. Here, we present a new implementation platform for bias adjustment, which we call MIdAS (MultI-scale bias AdjuStment). MIdAS is a modern code implementation that supports features such as: modern Python libraries that allow efficient processing of large data sets at computing clusters, state-of-the-art bias adjustment methods based on quantile mapping, "day-of-year" based adjustments to avoid artificial discontinuities, and also introduces cascade adjustment in time and space. The MIdAS platform has been set up such that it will continually support development of methods aimed towards higher resolution climate model data, explicitly targeting cases where there is a scale mismatch between data sets. The paper presents a comparison of different quantile mapping based bias adjustment methods and the subsequently chosen code implementation for MIdAS. A current recommended setup of the MIdAS bias adjustment is presented and evaluated in a pseudo-reference setup for regions around the world. Special focus is put on preservation of trends in future climate projections, and it is shown that the cascade adjustments perform better than the standard quantile mapping implementations, and often similar to methods that explicitly preserve trends.

## 1  Introduction

Bias adjustment is commonly applied to adjust results from climate models to make them compatible with impact models and for calculations of climate indicators (e.g. Teutschbein and Seibert, 2012; Maraun et al., 2017). The issue with using climate model data directly arises from systematic deviations at regional and seasonal scales in climate models compared to observations. The core of a bias adjustment is therefore an algorithm that transforms the model values toward a reference. Because typical bias adjustment tools are pure statistical processors that can reshape almost any timeseries to look like the target reference (Maraun et al., 2017), it is important to be aware of side effects of the adjustment.

The mythological king Midas wished for his touch to become a powerful transfer function that converts any physical object to gold. Indeed, he was granted this wish, and his touch was the first great example of bias adjustment.

*Gold! glorious gold! I am made up of gold!*

*I pluck a rose, a silly, fading rose,*

*Its soft, pink petals change to yellow gold;*

*Its stem, its leaves are gold–and what before*

*Was fit for a poor peasant's festal dress*

*May now adorn a Queen.*

*– Shelley (1922)*

Of course, he soon realized his folly and narrow minded reference, which although useful for one purpose, was devastating for many other.

*...this meat*

*which by its scent quickened my appetite*

*Has lost its scent, its taste,–'tis useless gold.*

*Alas! my fate! 'tis gold! this peach is gold*

*This bread, these grapes & all I touch! this meat*

*Which by its scent quickened my appetite*

*Has lost its scent, its taste,–'tis useless gold.*

*– Shelley (1922)*

King Midas' pioneering career as a bias adjuster quickly fell apart after this initial failure. Thousands of years and another reality later, bias adjustment was picked up by the climate community. Several decades of step-wise improvements have sig-
45 nificantly advanced the research field. From the first tools of mean bias adjustment through the delta change approach (e.g. Gleick, 1986), methods have evolved to more complex methods that account for complete distributions of a variable (e.g. Wood et al., 2002; Piani et al., 2010), multi-variate features (Piani and Haerter, 2012; Vrac and Friederichs, 2014; Francois et al., 2020), temporal resolution (Haerter et al., 2011; Johnson and Sharma, 2012; Mehrota and Sharma, 2015; Nguyen et al., 2016), spatial mismatch with observations (Haerter et al., 2015), to name some advancements. Adjustments are often applied to
50 smaller regions, e.g. for a local impact model, where the range of values and climate regimes is constrained. When employing bias adjustment across the globe, the methods have been exposed to extreme conditions that they might not have been developed or evaluated for, such as very dry climates and regions of strong orography (Pechlivanidis et al., 2016; Photiadou et al., 2021). Further, a global reference data set can vary greatly in quality across the world, mainly as a consequence of available observational data (Berg et al., 2021a; Hassler and Lauer, 2021).

The story of king Midas also teaches us to be aware of side-effects of the bias adjustment on scales beyond our main focus. Haerter et al. (2011) identified the potential interaction between statistics at different temporal time scales. They introduced the concept of "cascade bias adjustment" by separately applying bias adjustment to monthly mean data and daily anomalies, which are later merged to form the final adjusted timeseries. The motivation for the cascade method is to avoid introducing bias in one temporal resolution while adjusting another, which can occur in cases with bias in variance. The spatial character of the

climate model and the observation was explored by Berg et al. (2015) for situations where the climate model has finer resolution than the reference. They presented a method where a "pseudo-reference" was produced by merging model and observations, such that coarse scales agree with the reference and the finer spatial anomalies are added to this coarse background. The two approaches can be connected by applying a spatial cascade, where a coarse spatial resolution and finer scale anomalies are adjusted separately, or only one of the scales is adjusted. This can be useful in cases where the observational reference is of coarser resolution than the model; something which will likely become more common as model resolution increases. However, currently the dominant case is that of combining bias adjustment and statistical downscaling in one single step (Teutschbein and Seibert, 2012; Fiddes et al., 2022). In this case it is also important to consider how the bias adjustment affects the statistics, e.g. considering the variance inflation issue (Maraun, 2013).

Another side-effect of bias adjustment can be a modulation of the climate change signal. This can be of a statistical nature, such as the non-physically justifiable scaling effect with a variance bias discussed by Haerter et al. (2011) and others (Themessl et al., 2011; Berg et al., 2012). However, there may also be physical reasoning for bias adjustment improving certain processes in the model which could justify a modulation of the climate change signal (Buser et al., 2009; Boberg and Christensen, 2012), although this is far from trivial (Maraun, 2016).

Here we present a new take on king Midas attempts by introducing MIdAS (MultI-scale bias AdjuStment). MIdAS is a bias adjustment software based on empirical quantile mapping, which for the first time introduces cascade adjustment in time and space, as an option to standard bias adjustment. Further, to avoid artificial discontinuities between calendar months, MIdAS makes use of day-of-year scaling steps. To balance the increased computational costs that comes with the day-of-year and multiple cascades, a significant effort has been put into multi-processing methods to allow scaling of the calculations on large computational clusters. MIdAS is therefore coded in Python 3, using libraries well adapted to large computing clusters and automatic parallelization. The emphasis on large-scale application of the code separates MIdAS from most earlier published bias adjustment codes with focus mainly on presenting the method, and not on the practical usage.

In this paper, we present MIdAS with its main methods, assumptions and configurations, as well as a comprehensive evaluation and model inter-comparison of MIdAS and other state-of-the-art methods, including several trend preserving methods. Using a pseudo-reality setup with known modelled future pseudo-observations, the impact of the different bias adjustment methods are compared for various statistics. We especially address the cascade adjustments for time and space, and for different variables, and compare with published quantile mapping (QM) methods. We target non-parametric methods so that we can evaluate the cascade option for all methods, and focus on the most often applied and cited methods.

## 2 Bias adjustment methods

### 2.1 MIdAS method description

The core of the MIdAS method is the bias adjustment step, which is described in Section 2.1.1. However, there are necessary preparations of data before and after that step, which demands careful and quality assured processing. This aspect is of increasing concern as domain and ensemble sizes of climate model data put higher demand on the performance and scalabil-

ity of the bias adjustment software. For these reasons, MIdAS has been developed with modern computing tools, including parallelization and input-output handling, as described in Section 2.1.2.

### 2.1.1 Theory

The core method of bias adjustment in MIdAS is an empirical quantile mapping (EQM). The basic requirements are a reference timeseries $x$, typically from observations, and a model timeseries $y$. A sub-period is defined for calibrating the bias adjustment, e.g. 1971–2000. Because biases often differ between seasons, the timeseries are split in smaller sections depending on the time of the year. Most methods do this by calendar month, but in MIdAS we opt for an approach where a window around each day of the year ($doy = [1, 365]$) is chosen. The standard setting is to use 15 $d$ before and after, such that 31 $d$ are used to build the distribution of the reference and model data, specifically for each $doy$. Leap days are not explicitly handled, and the $doy$ is defined from the first of January and counts the days until 365 is reached. Therefore, the $31^{st}$ of December on leap years is not included, but will be bias adjusted with the same parameters as $doy = 365$.

In standard QM, the cumulative distribution functions (CDFs) of the reference dataset $F_{x,doy(i)}$ and of the model dataset $F_{y,doy(i)}$ are used to calculate the bias adjusted value $z_i$ from the original value $y_i$ according to Eq. 1.

$$
\begin{aligned}
z_i &= F^{-1}_{x,doy(i)}(F_{y,doy(i)}(y_i)) \\
&= (F^{-1}_{x,doy(i)} \cdot F_{y,doy(i)})(y_i)
\end{aligned}
\tag{1}
$$

In practice, the CDFs are unknown and must be approximated. For EQM, this is done by the empirical cumulative distributions functions with linear interpolation between neighboring data points to avoid unphysical jumps. Since the number of data points defining the approximated CDFs is the same for reference and model due to our selection of points, we can pair them to form the points in the so-called Q-Q plot, which sorts the data in ascending order and plots them against each other, see Fig. 1. If we perform linear interpolation between these points and call the resulting function $f$, we see that this is identical to the EQM approach, i.e.

$$
F^{-1}_{x,doy(i)} \cdot F_{y,doy(i)} = f_{doy(i)}.
\tag{2}
$$

The main downside of the EQM approach is its use of resources. Due to the relatively large number of points, a commensurate amount of storage is needed because all points of the calibration dataset must be stored as parameters for the bias adjustment. Furthermore, the evaluation of the adjustment function becomes more costly as more points (longer reference period, longer time window) are considered since the relevant pair of data points must be located first. This is done to diminishing returns because generally speaking, the vast majority of points falls into the same central interval, doing little to improve the quality of the adjustment.

Our goal with MIdAS is to find a good approximation for $f$ that does not suffer from these problems and offers better resource efficiency and scalability. We achieve this by fitting a linear smoothing spline function to the Q-Q plot (similarly to the fitQmapSSPLIN function proposed by Gudmundsson et al. (2012)), using the routine splrep from Virtanen et al. (2020), which in turn is based on the FITPACK routines by Paul Dierckx (Dierckx, 1975, 1981, 1982, 1995). This approach allows a

good approximation of $f$ with far fewer knots in the spline, while still guaranteeing a good representation of all data points. It is tempting to go to higher order, e.g. cubic, splines to achieve an even better and smoother representation, possibly with even fewer knots. However, experience shows that this can introduce overshooting behavior, particularly at the ends of the interval, whereas linear splines give a faithful reflection of the calibration data (Piani and Haerter, 2012). The spline fitting function also provides us with the possibility of using weights for individual data points. We use this to mitigate the following problem.

Earlier studies have shown the sensitivity of such EQM approaches regarding the tail behaviour (Switanek et al., 2017). To avoid excessive impact from outliers in the tails, a linear function is fitted to the 90% most central data points of the Q-Q plot. The weights to the spline function are then defined according to the standard deviation of the data points from the linear fit. To avoid excessive weights for individual points, those points with a lower standard deviation than a threshold minimal standard deviation of 1% of the midpoint of all reference values, are set to the threshold. When the splines have been fitted for each $doy$, each data point of the model timeseries will be adjusted according to its value and $doy$ according to:

$$z_i = f_{doy(i)}(y_i), \tag{3}$$

where $z_i$ is the bias adjusted data for time step $i$. The scaling is as also outlined in Fig. 2.

In the case of a time cascade adjustment, we follow the method as outlined in Haerter et al. (2011) and extend it to include also a multiplicative case. Each timeseries is first split up in two separate timeseries:

$$x_i = x_i' + \overline{x_i}, \tag{4}$$

where $i$ is the timestep in days, and the two cascades are indicated by an overbar for the coarser temporal aggregation, e.g. $\overline{x_i} = \sum_{k=i}^{i+N-1} x_k/N$, where the default setting is $N = 31$, and a prime for anomalies thereof. The model timeseries, $y$ is split up in the same way. Equation 4 is applied to non-bounded variables such as temperature that can handle such additive splitting. For bounded variables, such as precipitation, the separation is instead multiplicative:

$$x_i = x_i' \cdot \overline{x_i}. \tag{5}$$

One can now chose to separately adjust each of the cascades, and then return the final bias adjusted timeseries by substituting for $z$ in Eq 4.

The spatial cascade follows similar form as the temporal, but is based on a coarse and a finer scale spatial resolution, with an additive or multiplicative operator as for the temporal cascades. Albeit not yet implemented in MIdAS itself, experiments using a spatial cascade have been performed in a so far not published research project in the Mashreq area where the obser-vational network did not support the higher regional climate model resolution. The spatial cascade was then using pre- and postprocessing to construct a coarse 25 km scale that was adjusted, and a finer scale of anomalies that were kept intact.

We are here addressing bias adjustment of daily mean temperature and precipitation. Whereas temperature follows the steps outlined above, an additional step is introduced for precipitation to adjust the number of wet days. MIdAS employs the SSR (Singularity Stochastic Removal) method for wet day adjustments (Vrac et al., 2016). SSR works in four steps:

1. Find the threshold lowest precipitation value, $P_{th}$, greater than zero across both timeseries.

2. Set all zero values to a random number in the range $(0, P_{th})$.

3. Let the bias adjustment step assign new values to the timeseries, including the values promoted from zero in step 2.

4. Finally, all values below $P_{th}$ are set to zero.

Together with the quantile mapping, SSR will ensure that the number of wet days are close to the reference data both when the bias is wet and dry. However, any removed excessive, or promoted dry, time step will hold no physical meaning in themselves. This is still a reasonable methodology since these values are normally only affecting the lower end of the precipitation distribution. Berg et al. (2021b) evaluated the SSR method for MIdAS for the same setup of GCM and domains as used in this paper. As reference, they used a simpler method where only excessive rain days (precipitation below 0.1 mm/day) were adjusted. In conclusion, they noted that the SSR outperformed the simpler method on all accounts, and especially for cases where the model underestimates the number of wet days where the simpler method can even increase the original bias.

### 2.1.2 Implementation

MIdAS is written in Python and makes use of its scientific software stack. Two libraries warrant special mention in the context of MIdAS as an extensible, flexible platform for the development of bias adjustment methods. We use Iris (Iris , 2021) to read and write netCDF files as well as for the handling of metadata such as units and calendars throughout the processing chain, and we use Dask (Dask , 2016) for the efficient in-core and out-of-core processing in a highly parallel computing environment.

Iris itself uses Dask as the backend for its data management, which makes the interaction with both packages seamless. There are other packages that support the reading of netCDF files, notably the netCDF4 Python packages and xarray. Of these, netCDF4 operates on a lower level with no knowledge about the applicable metadata conventions on climate data (the CF Conventions) and should generally be avoided for immediate use; both Iris and xarray rely on it for their support of netCDF files. Xarray is a serious contender and potential replacement for Iris. Iris was chosen for two main reasons. One is its better support of climate data and metadata standards. This stems from its specialization on this field, whereas xarray aims to be more widely applicable and is indeed used in applications far beyond climate. The other is familiarity of the authors, that have worked with Iris in other projects. Nevertheless, since both are underpinned by Dask, and the heavy lifting in MIdAS is carried out directly via Dask, a version of MIdAS with xarray would be conceivable. Using a standard library for the interaction with the data also prepares us for possible future changes, such as the use of other storage format like zarr on object stores, or the use of unstructured grids in future climate models.

Dask is a flexible library for parallel computing in Python. In particular, it allows for efficient parallelization that is mostly transparent from the application layer, yet can distribute the workloads to parallel computers — from modern laptops, that often are already equipped with several CPUs, to state-of-the-art High-Performance Computers (HPC), spanning many nodes and including GPUs. This flexibility allows us to tackle data volumes that are out of reach for more traditional programs, yet are becoming more and more common in Earth sciences as the resolution and complexity of models and other data sources is increased. It is worth pointing out, that often it is no longer computing capability per se, i.e. FLOPS, which is the limiting factor, but rather memory and bandwidth requirements. This, too, can be addressed by the use of several nodes in an HPC.

In general, MIdAS can be thought of in stages. First, we perform data input and metadata validation. Thanks to the concept of lazy data, this involves reading metadata from the disk, but *not* reading the actual data itself. This concept allows setting up and prepare all calculation steps before any data is read into memory, which allows efficient parallelization of both reading and calculations, with reduced bandwidth and memory requirements. Next, we perform the cascade separation step on the input timeseries, which results in a new set of timeseries. On each of these timeseries separately, we execute first the calibration, then the adjustment of the bias correction. Finally, the two timeseries are combined again into a single timeseries of corrected data and written to disk.

The flexibility of MIdAS comes from the structure of the program that allows an easy exchange of individual components of this chain. For example, if one wants to base the correction not on day of year, but rather on months, only the implementation of the aggregation of the order statistics needs to be changed; all other aspects of the chain can remain untouched. If one wants to perform a parametric correction instead of the spline based approach presented here, only the so-called kernel will need to be exchanged. Likewise, different cascading approaches can be combined with the underlying correction methods. All the while, the benefit of high efficiency and parallelization can be maintained with little impact even on the implementation of new bias correction methods. Making this flexibility available to the user in a simple configuration approach is one of the near-term goals of further MIdAS development.

## 2.2 Additional methods for bias adjustment

To put MIdAS' performance in context, other often cited and used methods from different flavours of QM are included in the evaluation (see also Fig. 2). The included methods are of an empirical nature, and we have on purpose left out parametric methods such as ISIMIP3 (Lange, 2019) and DBS (Yang et al., 2010) due to difficulties in identifying the appropriate distribution functions for different cascades. So although parametric methods have value for many applications, they are out of scope for the analyses performed in this paper.

**qmapQ** EQM considers each single data point in $F$, which works well in the thicker centre of the distributions. However, the methods becomes more sensitive at the tails. Switanek et al. (2017) showed that if the sensitivity to outliers in the tails is not addressed, it can have severe impacts when applied in another climate period. This issue has been dealt with in different ways, and here we used the method we call "qmapQ" from (Gudmundsson et al., 2012), which divides the data range in one hundred quantile steps, which ensures equal number of data points in each sample. Both temperature and precipitation are treated equally. The correction found for the highest quantile is used to estimate those larger than the training values. We use the qmap library in R (Gudmundsson, 2016).

**DQM** by Cannon et al. (2015), performs a first step of detrending the timeseries for the mean value. An EQM is then applied, after which the trends are added to the bias adjusted data. The trends are calculated as discrete differences between 30-year time slices of the models for a future and a historical reference period, see Section 2.3.

**QDM** by Cannon et al. (2015) follows the same basic principles as EQM, but instead of initiating the transformation by the value, it originates from the quantile value of $F$. This gives the same result as EQM for the reference period, but may

differ for other periods. Here, we calculate the future distribution for set time-slices of the investigated periods. QDM further removes a linear trend before the bias adjustment step, and adds it back afterwards, in order to retain the original climate signal.

**CDF-t** by Michelangeli et al. (2009) is a more intricate method which produces an estimation of the future reference distribution, which is then applied for the adjustment. In this way, it reduces the dependency of the stationarity assumption of most bias adjustment methods. In a first step, an adjustment is made at the quantile level, as in QDM, and a pseudo-future observation is constructed. The implementation in the R-package "CDFt" is used (Vrac and Michelangeli, 2009).

### 2.3 Evaluation scheme

We make use of the so-called pseudo-reality approach of evaluating the bias adjustment methods (Maraun, 2012; Räisänen and Räty, 2012; Räty et al., 2014; Schmith et al., 2021). In pseudo-reality, a model ensemble with a historical and future projection is employed and the reference will not be an observational data set, but instead one of the ensemble members. This has the advantage of allowing the models to be assessed for the past as well as the future climate that is the target of the investigations. Each model will in turn be given the role of the pseudo-observations, i.e. to act as the reference for the bias adjustment of

the other models. Calibration of the bias adjustment methods is performed for the period 1971-2000, and then applied to the periods 1971-2000, 2011-2040, 2041-2070, and 2071-2100. The bias is then evaluated for each of the periods, with the first being the calibration period with the expected best performance, and the latter three acting as validation periods. Further, the latter three periods allows assessment of the impact of bias adjustment on the climate change signals.

### 2.4 Evaluation parameters and ranking

All statistics are calculated for each calendar month, and across 30-year time-slices. The statistic operators, $O$, include: the mean, the 0, 1, 99 and $100^{th}$ percentiles (referred to as min, 1p, 99p and max, respectively), the number of wet days (for precipitation), and the PDF (probability density function) skill score (pdfSS). pdfSS is defined as the overlapping area of two PDFs, which leads to a value 0 for no overlap to 1 for a perfect overlap. The bias, $\beta$ is calculated for all but pdfSS, for each month $m$, climate model $i$ and grid point $g$ in a domain:

$$\beta_g(m,i) = O_{mod,g}(m,i) - O_{ref,g}(m). \tag{6}$$

The bias is summarized per domain (averaging operator "$<>_z$") as an absolute bias:

$$\beta = < |\beta_g(m,i)| >_g \tag{7}$$

One can expect $\beta$ to be close to zero in the calibration period, but is likely non-zero for other periods. A basic assumption of the bias adjustment methods is that of time-stationarity in model bias, which means that the remaining bias outside the

250 calibration period should preferably be low and near constant. The impact of the bias adjustment on the climate change signals, $\Delta$ is investigated similarly as the bias:

$$\Delta_g = O_{mod,g}(m,i) - O_{ref,g}(m,i) \tag{8}$$

**Table 1.** Description of the ranking methods.

| Method | Description |
|---|---|
| 1 | Ranking across all ensemble members, regions, periods and months. |
| 2 | Ranking for ensemble mean of all ensemble members, averaged over all periods and months. Then the ranks are averaged across all domains. |
| 3 | Ranking for ensemble median, averaged over all periods and months. Then the ranks are averaged across all domains. |

and

$$\Delta = <\Delta_g(m,i)>_g . \tag{9}$$

Three different ranking methods are then calculated, see Tab. 1. The statistics entering the ranking can be $\beta$ for the different statistics (low values for good performance, except pdfSS for which high values marks good performance), or the modification of the climate change signal $\Delta$. For the latter, one often strives to reduce impacts of the bias adjustment on the climate change signal (Maraun, 2016; Ivanov et al., 2018; Casanueva et al., 2018). However, through detailed studies of how a model through a poorly simulated physical process achieves bias, one can justify the bias adjustment method to affect the change signal by

improving the model. We do not go into such depth in the current evaluation.

## 3   Data

Daily mean temperature and precipitation from four global climate models (GCMs) with the RCP8.5 emission scenario are included in the study, see Tab. 2. The concept of model genealogy (Knutti et al., 2013), i.e. to which degree models are related in terms of e.g. parametrizations, was used to make a sub-selection of models. From an ensemble of opportunity of 18 GCMs,

four models were chosen such that they were as far apart as possible in the mapping of Knutti et al. (2013). It was assumed that the further away the models are according to the genealogy, the more independent they are. Model independence is a desired feature in a pseudo-reality experiment.

    The evaluation is for computational reasons limited to a selection of ten regions around the world, see Fig. 3. The regions were chosen based on personal experience with previous applications of bias adjustment with issues such as inflation of variance

(NAM_1), dry regions (AFR_1), heavy precipitation (AFR_2, SAM_2), monsoon (WAS_1), strong land-sea contrasts (EAS_1, SAM_1) different challenging climates for the models (AUS_1, ARC_1), or simply locations of particular interest (EUR_1).

**Table 2.** List of the CMIP-5 GCMs included in the evaluation along with their RIP (realisation-initialization-physics) code. All GCMs were first remapped to a common 2.0 degree regular longitude-latitude grid.

| GCM | RIP |
| --- | --- |
| NorESM1-M | r1i1p1 |
| IPSL-IPSL-CM5A-MR | r1i1p1 |
| Inmcm4 | r1i1p1 |
| MPI-M-MPI-ESM-MR | r1i1p1 |

## 4 Results

### 4.1 Temporal cascade adjustments

In a first study, the temporal cascade is investigated for three different methods: QDM, DQM, and qmapQ. The SSR method was employed to adjust wet days for all methods, and in the cascade case it is applied before the timeseries is split into the cascades. Each model performs two experiments: a baseline without cascade adjustment, and an experiment including two cascades at 31-day coarse temporal resolution and its daily anomalies. Figures 4 and 5 show the resulting median bias across all ensemble members in the pseudo-reality setup, with the different geographical regions on the vertical axis, and separately for each month of the year on the horizontal axis. Bias is normalised per regions because the bias can differ substantially in some cases.

For precipitation (Fig. 4), qmapQ and qmapQ_casc consistently outperform the other methods for all regions and all rankings. DQM performs generally well, but suffers from some outlier bias for certain regions and months. We did not go into closer evaluation of the reasons, other than verifying that the method is correctly implemented and that the issues lie with the methods themselves. Although not seen in the presented statistics, qmapQ_casc can for some regions and seasons lead to strong remaining bias compared to the other methods. A likely cause is the possibility of having large multiplicative factors when two small values are compared. Because bias adjustment is often part of a process chain to achieve an assessment of impact of climate change, such uncertainty in the results is problematic as it may give rise to issues downstream in the production.

Figure 5 shows that all methods succeed in substantially reducing the bias for mean temperature. QDM leads in all three ranking methods, followed closely by QDM_casc. Thereafter follows qmapQ and DQM, with qmapQ_casc consistently outperforming the baseline qmapQ. The cascade option is deteriorating the results for DQM and QDM.

In conclusion, the evaluation of the cascades have shown promise for both variables with the qmapQ method, but less so for the other bias adjustment methods. Further work on constraining the cascades for precipitation is needed, and we therefore decided to proceed with using a temporal cascade only for temperature, whereas precipitation adjustments are based on the original daily data. Temperature is also the variable with the strongest increase as a result of climate change, and therefore with the most likely potential of destructive interference between the temporal scales (Haerter et al., 2011).

## 4.2 Method inter-comparison

Based on the initial results on the cascade method presented in Section 4.1, the MIdAS bias adjustment method was implemented using a cascade of 31-day means and daily anomalies for temperature, and a single absolute daily data adjustment for precipitation. Here, MIdAS is compared with the methods listed in Section 2.2, with each method in its standard settings,

except for qmapQ which is accompanied by the cascade version qmapQ_casc used in Section 4.1. It should be noted upfront that MIdAS differs from the other methods by the running window approach, compared to other methods' discrete steps on calendar months when calculating the transfer functions. The implication is that MIdAS will suffer in the calendar month-based statistical analysis, which should be kept in mind when evaluating its performance. All methods are implemented using the SSR method for wet day adjustments, with exception of the CDF-t method, which has its own built-in adjustment.

The pseudo-reality evaluation is performed through analysis of summary statistics as presented in Fig. 6–9. Each figure presents an overview of the different analysed regions, for all time periods, and all calendar months of the year, and for each bias adjustment method. Further, each plot is accompanied by the ranking scores, where the summary ranking 1 (see Section 2.4) is presented for all statistics in Tab. 3 and Tab. 4. The data are normalised per region for presentation clarity, and presented separately for bias, $\beta$, and impact on the climate change signal, $\Delta$.

### 4.2.1 Precipitation

Figure 6 shows the results on bias for mean precipitation. The most striking feature of this figure is that all methods generally succeed in reducing the original bias of the models (leftmost column). However, there are some exceptions where different methods fail, and even increase the bias, e.g. CDF-t in AFR_1 and SAM_1, and DQM in AFR_1. The rankings indicate qmapQ, qmapQ_casc and MIdAS as top three performers, with QDM advancing before MIdAS for ranking 3. These four

methods perform rather similarly to each other.

Table 3 presents a summary of ranking 1 for all statistics. MIdAS is always in top three of the methods, with scores close to the better ranking methods. Considering that MIdAS suffers from its running mean temporal window in this comparison, its performance can be considered to be at least on par with qmapQ, qmapQ_casc and QDM. Notably, MIdAS's performance for maximum precipitation is high, whereas the otherwise top scoring methods qmapQ_casc and QDM perform worse.

The impact of the different methods on the climate change signal is presented for mean precipitation in Fig. 7. By visual inspection, MIdAS performs similarly to qmapQ and qmapQ_casc. Some more prominent differences are visible for AFR_2 in the beginning of the year. Whereas most methods have a generally amplifying impact on the climate change signal, the CDF-t method is reducing the magnitude of the changes. DQM has generally lower impacts, but also suffers from producing outliers in some cases, which impact strongly on the ranking statistics. The top three ranking methods, i.e. those with the smallest

impact on the change signals, are qmapQ_casc, CDF-t and qmapQ. In forth place is MIdAS, followed by the QDM and DQM methods. It is interesting that the trend preserving methods QDM and DQM perform worse in this comparison.

**Table 3.** Bias as calculated in ranking 1, across all regions, time periods and months. Occasions where the statistic was strongly affected by outliers is marked "inf", for infinity. The units are in mm d$^{-1}$ for precipitation, and $^{\circ}$C for temperature, except for the pdfSS statistic which is unitless.

| Variable(statistic) | uncorr | MIdAS | qmapQ_casc | qmapQ | QDM | CDF-t | DQM |
|---|---|---|---|---|---|---|---|
| P(min) | 0.08 | 0.04 | 0.04 | 0.04 | 0.04 | 0.04 | inf |
| P(mean) | 1.39 | 0.47 | 0.43 | 0.43 | 0.53 | 0.56 | inf |
| P(99p) | 7.99 | 3.10 | 4.14 | 3.08 | 3.27 | 3.66 | inf |
| P(max) | 17.69 | 10.20 | 14.31 | 9.41 | 11.12 | 10.43 | inf |
| P(pdfSS) | 0.75 | 0.89 | 0.89 | 0.89 | 0.88 | 0.81 | 0.87 |
| T(min) | 3.83 | 1.89 | 1.51 | 1.4 | 1.55 | 1.77 | 1.47 |
| T(1p) | 3.44 | 1.23 | 1.09 | 1.11 | 1.01 | 1.1 | 1.06 |
| T(mean) | 2.88 | 0.82 | 0.83 | 0.93 | 0.8 | 0.82 | 0.82 |
| T(99p) | 2.86 | 1.35 | 1.29 | 1.16 | 1.04 | 1.12 | 1.06 |
| T(max) | 2.96 | 1.76 | 1.6 | 1.26 | 1.23 | 2.69 | 1.15 |
| T(pdfSS) | 0.5 | 0.78 | 0.78 | 0.76 | 0.78 | 0.78 | 0.78 |

**Table 4.** Modulation of the climate change signal as calculated in ranking 1, across all regions, time periods and months. Occasions where the statistic was strongly affected by outliers is marked "inf", for infinity. The units are in mm d$^{-1}$ for precipitation, and $^{\circ}$C for temperature.

| Variable(statistic) | MIdAS | qmapQ_casc | qmapQ | QDM | CDF-t | DQM |
|---|---|---|---|---|---|---|
| P(min) | 0.07 | 0.04 | 0.04 | 0.03 | 0.46 | inf |
| P(mean) | 0.17 | 0.14 | 0.17 | 0.22 | 0.15 | inf |
| P(99p) | 1.49 | 1.32 | 1.6 | 1.17 | 1.16 | inf |
| P(max) | 2.60 | 3.31 | 1.41 | 5.22 | 2.32 | inf |
| T(min) | 0.67 | 0.7 | 0.99 | 0.07 | 0.83 | 0.53 |
| T(1p) | 0.54 | 0.57 | 0.79 | 0.03 | 0.35 | 0.25 |
| T(mean) | 0.42 | 0.43 | 0.58 | 0 | 0.1 | 0 |
| T(99p) | 0.41 | 0.33 | 0.42 | 0.02 | 0.26 | 0.21 |
| T(max) | 0.51 | 0.33 | 0.02 | 0.17 | 2.24 | 0.66 |

The results are fairly consistent with the above conclusions across the other statistics, as shown for ranking 1 in Tab. 4. The main differences is the better performance of QDM for minimum and 99p precipitation, and the better performance of MIdAS for 99p and maximum precipitation.

### 4.2.2  Temperature

Figure 8 shows that the mean temperature is, like mean precipitation, well adjusted by all methods. The only strong outstanding features are from qmapQ and to a lesser degree midas in AFR_1, and qmapQ_casc at the end of the year in ARC_1. The late autumn in ARC_1 is indeed challenging for all methods, with remaining bias similar to, or even higher, than the original bias. QDM is clearly outperforming all other methods, with top scores in each ranking. However, the differences are very small between QDM, CDF-t and MIdAS, whereas DQM and the qmapQ methods perform only slightly worse.

Ranking 1 in Tab. 3 shows that QDM is also the overall best method for the different statistics. The differences are, however, very small across the different methods - often between $0.1 – 0.5\ °C$, which can be considered small compared to the original bias of 2.9 to $3.8\ °C$.

Regarding the impact on the climate change signal, the initial detrending performed in QDM results in non-existing or very small impacts on the mean, see Fig. 9. Also DQM and CDF-t explicitly account for the climate change signal, and have low impacts on the signal for mean temperature. However, Tab. 4 reveals that the impact is stronger on other statistics that are not explicitly accounted for, most notably minimum and maximum temperature, which underlines the importance of evaluating many characteristics of the adjusted models (Maraun and Widmann, 2018). Again, QDM is keeping the impact on the climate change signal low and gets the overall best ranking, followed by DQM. CDF-t suffers from some larger outliers for minimum, and especially maximum temperature of several degrees Celsius. MIdAS, qmapQ, and qmapQ_casc are free to affect the change signal, but still remain at less than one degree Celsius, and often below $0.5\ °C$. qmapQ_casc has consistently lower impacts on the signal than qmapQ, which is expected (Haerter et al., 2011), and MIdAS is overall on par with qmapQ_casc.

### 4.3  Discussion

The idea behind the development of MIdAS was to primarily have a good platform to build bias adjustments methods on. The envisioned future applications of bias adjustment is on scales that reach beyond the resolution in both time and space of today's gridded observations. Therefore, it is necessary to be able to perform adjustments on resolutions where one have trust in the observations. As previously argued for by Berg et al. (2015), bias adjustment should avoid tampering with scales better simulated by the dynamic models. Many gridded data sets have different "models" for mapping (e.g. kriging or optimal interpolation), and may include effects such as orographic influence and wind exposure that affect precipitation. Such models are not necessarily better than the dynamical model's representation, and should therefore not obviously be imposed in a bias adjustment process. Spatial cascades can circumvent this issue, by e.g. only performing adjustment on a coarser scale. One example where MIdAS was applied in such a context was for a recent bias adjustment of downscaled CMIP6 data for the Mashreq domain. The reference observations used were at 25 km resolution, whereas the downscaled models were at 12.5 km resolution. A spatial cascade was then used to only adjust temperature at a coarsened spatial resolution, by employing a 3x3 grid point filter on the data, and then adding the original fine scale anomalies to the bias adjusted cascade. For precipitation, there are still some caveats to work out for the cascades. Because of the multiplicative "nature" of this zero-bounded variable,

the cascade can lead to exaggerated response when multiplying small numbers. It is therefore necessary to add constraints on the data.

Although it will likely become more common to have models with higher resolution than the reference data, the case is mostly the opposite today. Therefore, bias adjustment methods are often used to also downscale data by first remapping to the higher resolution of the reference data and then bias adjusting with an inherent scale adjustment. We have not yet evaluated whether a spatial cascade would affect the performance, but in this case the finer spatial scales need to be included in the processing.

MIdAS was developed with a perspective to evolve into multi-variate methods (see Francois et al. (2020) for a review). Such methods pose not only increased complexity to the bias adjustment algorithms, but also increased complexity and demand on the infrastructure of the code to handle the multiple variables. The cascade perspective of bias adjustment has been studied by Mehrota and Sharma (2015), which could form a starting point for this development.

The decision to use empirical quantile mapping in MIdAS is primarily based on the complexity in finding appropriate distribution functions for the different cascades. Fitting a spline with a free number of nodes will in the majority of cases reduce the distribution to a simpler form which does not overfit the spline to random deviations (i.e. noise) in the sample, which can be the case for point-by-point empirical functions. Another strong argument is the sometimes poor fit of the distributions (e.g. Gamma or Gaussian) for all points of the Earth, which might require additional post processing to give reasonable results. When applying bias adjustment as a production mode for a larger ensemble or for a climate service, frequent poor fits can cause severe disruptions in the production, reduced quality of the output data, and difficulties in transparently describing the methodology.

Most bias adjustment approaches are implemented to be applied by calendar month. This can introduce significant unphysical steps in the timeseries between different months. The day-of-year 31-day window used in MIdAS avoids this issue, but at a computational cost with more than 30 times the number of calculations for the calibration. The higher detail in this approach improves the bias adjustment. However, using standard evaluation on calendar months, the method will appear to perform worse as the results within a month are based on statistics overlapping parts of multiple calendar months. This must be kept in mind when performing inter-comparison evaluations of bias adjustment methods. Further, we note that the day-of-year approach will inherently lead to underestimation/overestimation of the bias-adjusted highest/lowest values. The reason is that the day-of-year approach applies the adjustment to a subset (i.e. the central day of the window) of the calibration data only. In quantile-mapping, the highest value in the calibration data for the model data is mapped to the highest value in the calibration data of the reference data (vice versa for lowest value). Dependent on the location of the window maxima and minima in the data, it can happen that the these extremes are never reached in the mapping. Note that this is only noticeable in the reference period, for which no extrapolation occurs. This is more pronounced for precipitation than temperature and linked to the tail-heavy characteristic of daily precipitation.

# 5 Conclusions

The MIdAS model for bias adjustment was presented. Currently, the core functions of bias adjustment is similar to other released EQM methods, and the implementation of MIdAS performs on par with or better than the selection of state-of-the-art methods included in the evaluation.

Further, MIdAS has the following additional features as compared to currently released bias adjustment software:

– cascade adjustments to separately bias adjust a coarse scale, and smaller scale anomalies. This is implemented for temporal cascades in the code, and can be performed by pre- and post-processing for spatial cascades as well.

– day-of-year running window to build the transfer function, rather than the standard calendar month discrete steps. This removes potential nonphysical steps between adjacent days in different months.

The processing platform of MIdAS has been constructed for adding features such as multi-cascade (time and space "scale") and multi-variate bias adjustment methods. This makes MIdAS a good base to build future development of bias adjustment on. For example, future developments of MIdAS will in the short term include extension to more variables such as wind speed and relative humidity, and consistent adjustment of minimum and maximum temperature. Also the user interface will be developed for easier setup of options and execution of the program. In the longer term, development will focus on introducing spatial cascades and multi-variate methods.

*Code and data availability.* The MIdAS git repository is open for all to access and use under the GNU LESSER GENERAL PUBLIC LICENSE v3, at https://git.smhi.se/midas/midas. We welcome participation in the further development of MIdAS by requesting developer access through the git repository.

The code for the presented version of MIdAS is available at https://doi.org/10.5281/zenodo.6624233 under the GNU LESSER GENERAL PUBLIC LICENSE v3 (Berg et al., 2022a). The repository includes documentation of the method, automatic set up scripts for a conda environment to run the model, and information on how to apply it. All climate model data used in the pseudo-reality experiments were downloaded from the ESGF (Earth System Grid Federation) nodes. The scripts for producing the inter-comparison of bias adjustment methods and the analysis are available from https://doi.org/10.5281/zenodo.6043222 (Berg et al., 2022b). The repository also contains the remapped excerpts of the global climate models and the resulting files from the bias adjustments that are presented in the paper.

*Author contributions.* PB was leading the method development, and contributed to the experiments. TB set up the experiment design and carried out the evaluation. WY implemented the method-intercomparison and carried out the experiments. KZ performed the MIdAS technical implementation and code design. PB prepared the manuscript, with contributions from all co-authors.

*Competing interests.* The authors declare that they have no conflicts of interest.

*Acknowledgements.* The development of MIdAS was supported by SMHI and the project DIRT-X. DIRT-X is part of AXIS, an ERA-NET initiated by JPI Climate, and funded by FFG Austria, BMBF Germany, FORMAS Sweden, NWO NL, RCN Norway with co-funding by the European Union (Grant No. 776608). We would especially like to thank our SMHI colleagues Lars Bärring, Yeshewatesfa Hundecha, Magnus Hieronymus, Marco Kupiainen, Grigory Nikulin, Elin Sjökvist and Renate Wilcke for useful discussions and inputs on the development of MIdAS. Further, we appreciate the detailed and useful feedback from the three reviewers Jorn Van de Velde, Faranak Tootoonchi, and Joel Fiddes.

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

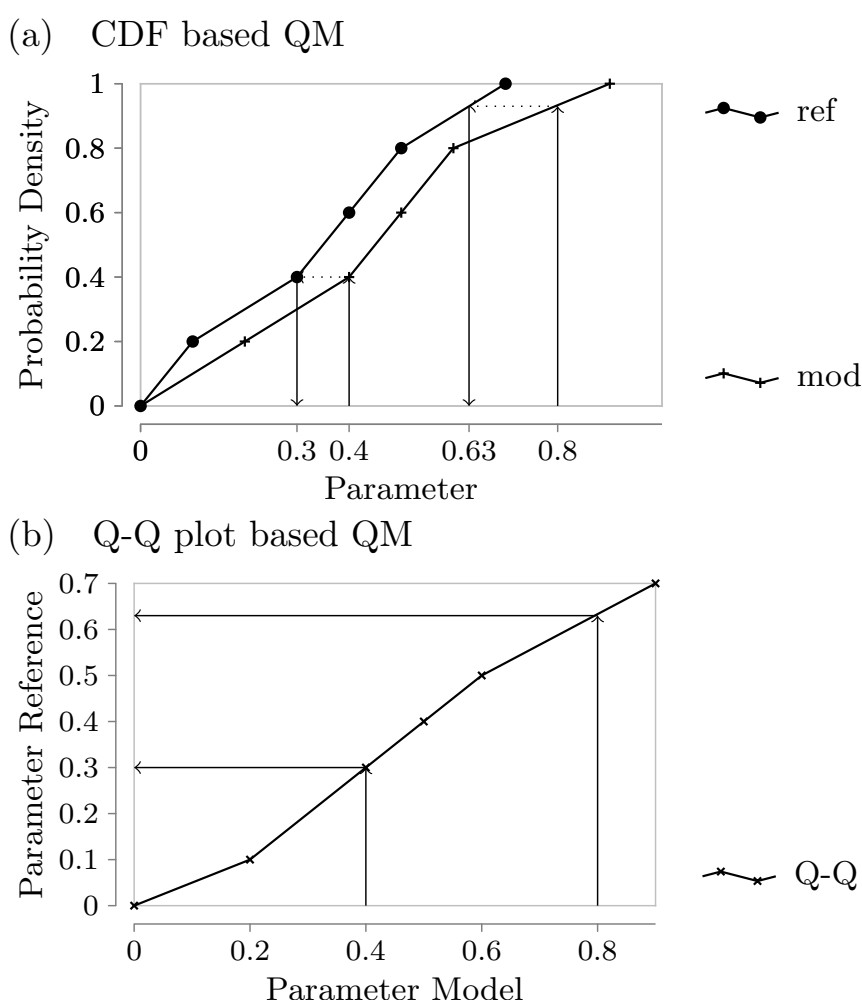

**Figure 1.** Illustration of the equivalence of CDF based EQM and Q-Q plot based EQM. In (a) we plot the linear interpolation version of the empirical CDFs for two example datasets, while (b) shows the Q-Q plot that results from the same data.

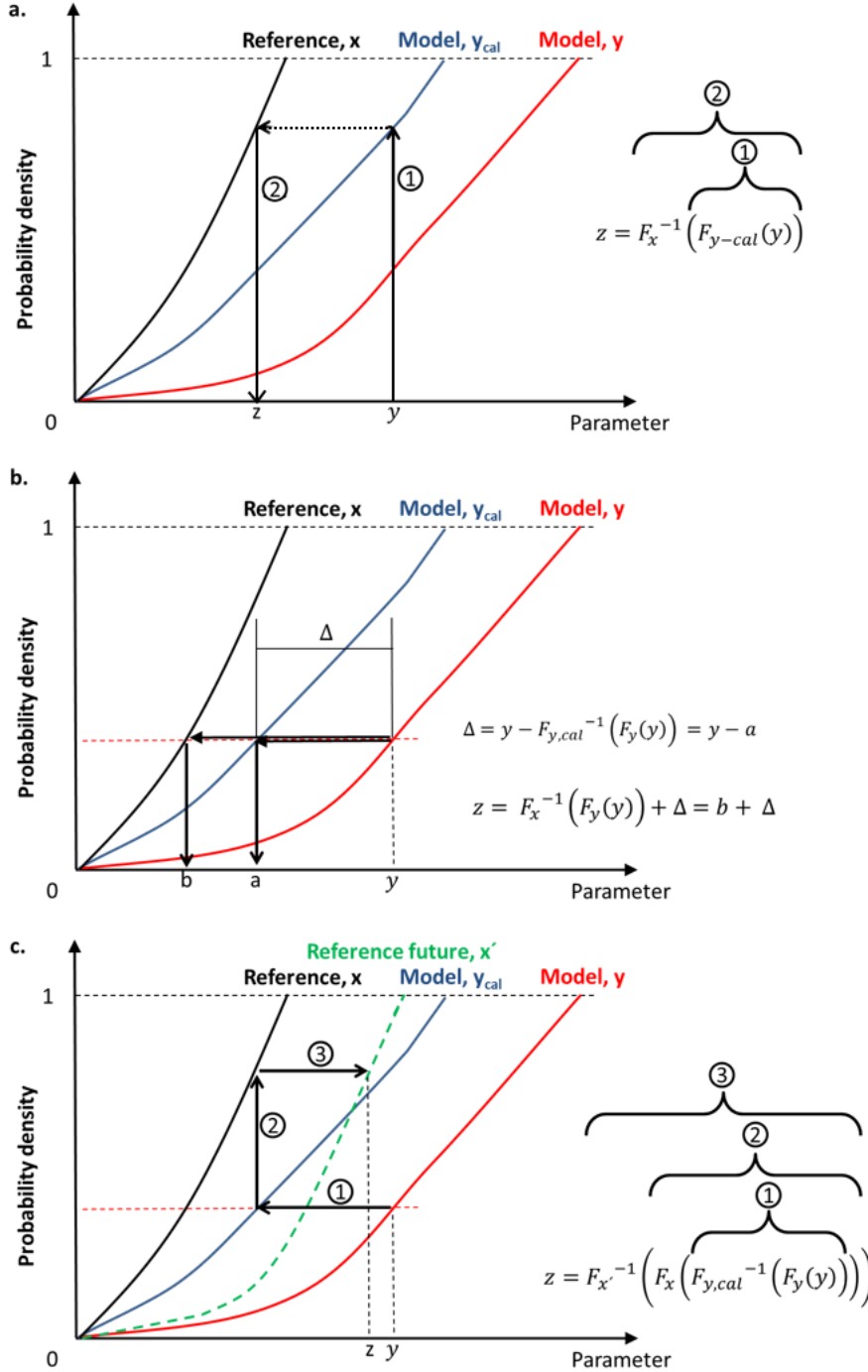

**Figure 2.** Descriptive visualisation of the quantile mapping for (a) MIdAS, EQM, and DQM, (b) QDM, and (c) CDF-t.

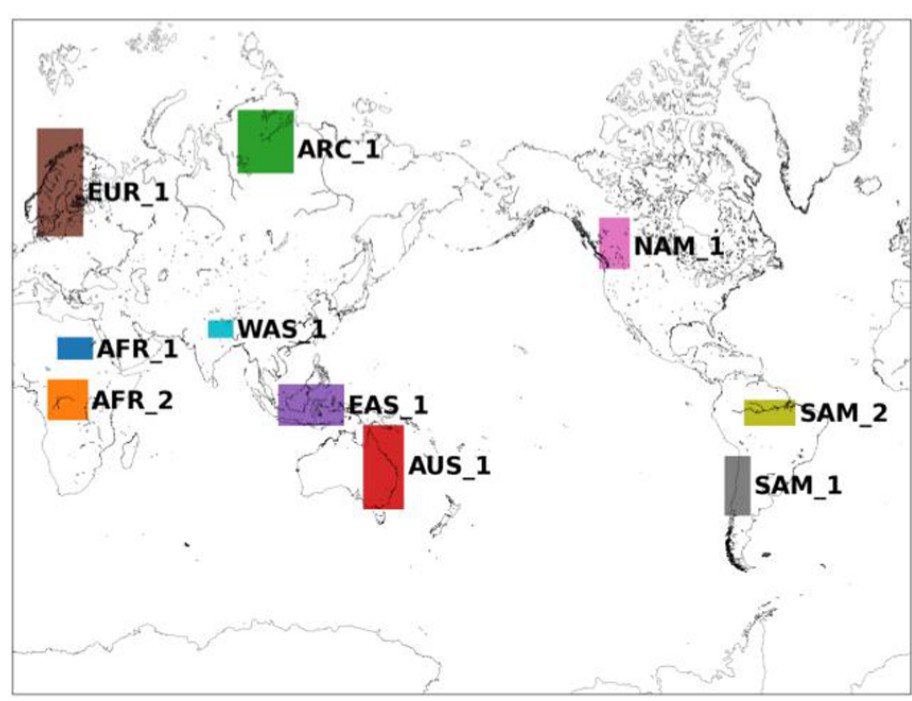

**Figure 3.** Evaluation domains.

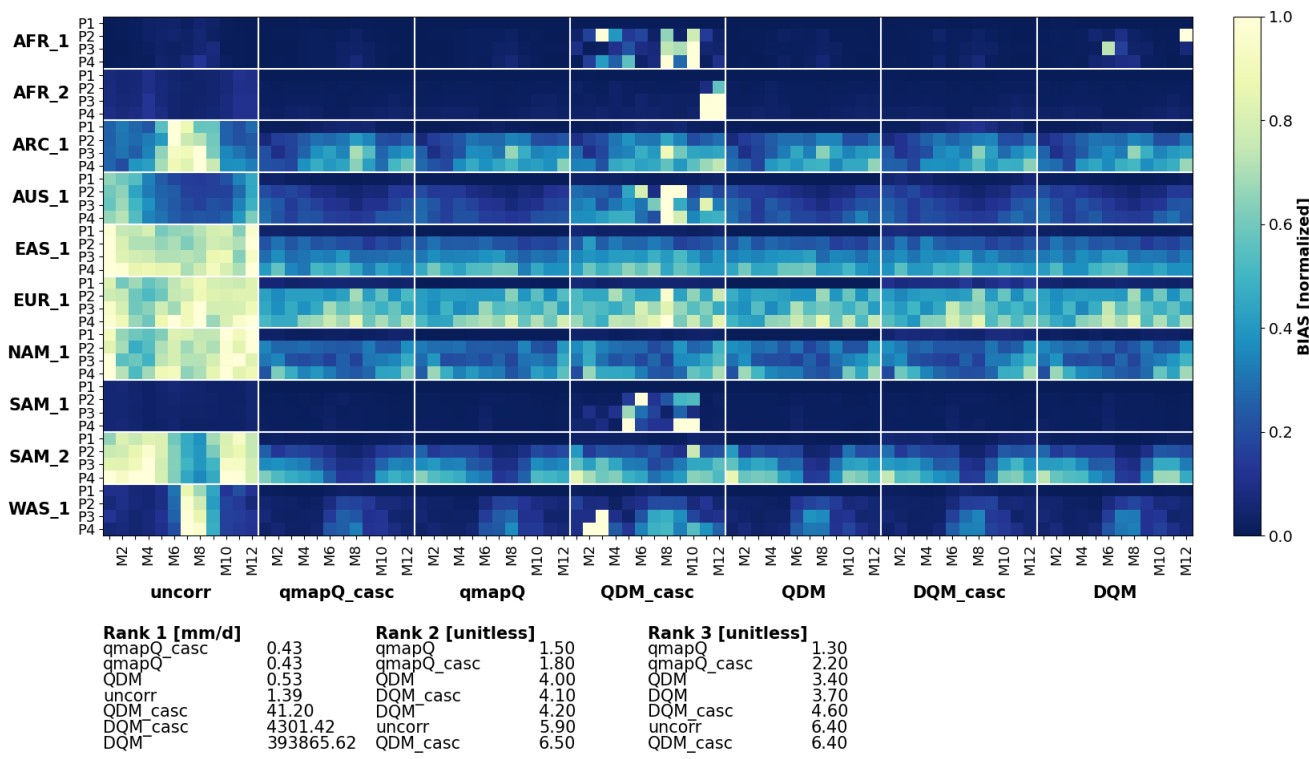

**Figure 4.** Bias of mean precipitation in the original data (uncorr) and after applying the different bias adjustment methods with (subscript "casc") and without temporal cascades. P1–4 mark the evaluation periods 1971-2000 (also calibration period), 2011-2040, 2041-2070, 2071-2100. The results are presented for each of the domains (vertical) and for each month of the year (M1–12). The results are normalised for each sub-panel, i.e. each domain, such that the bias of the uncorrected data and all methods are in the range 0–1.

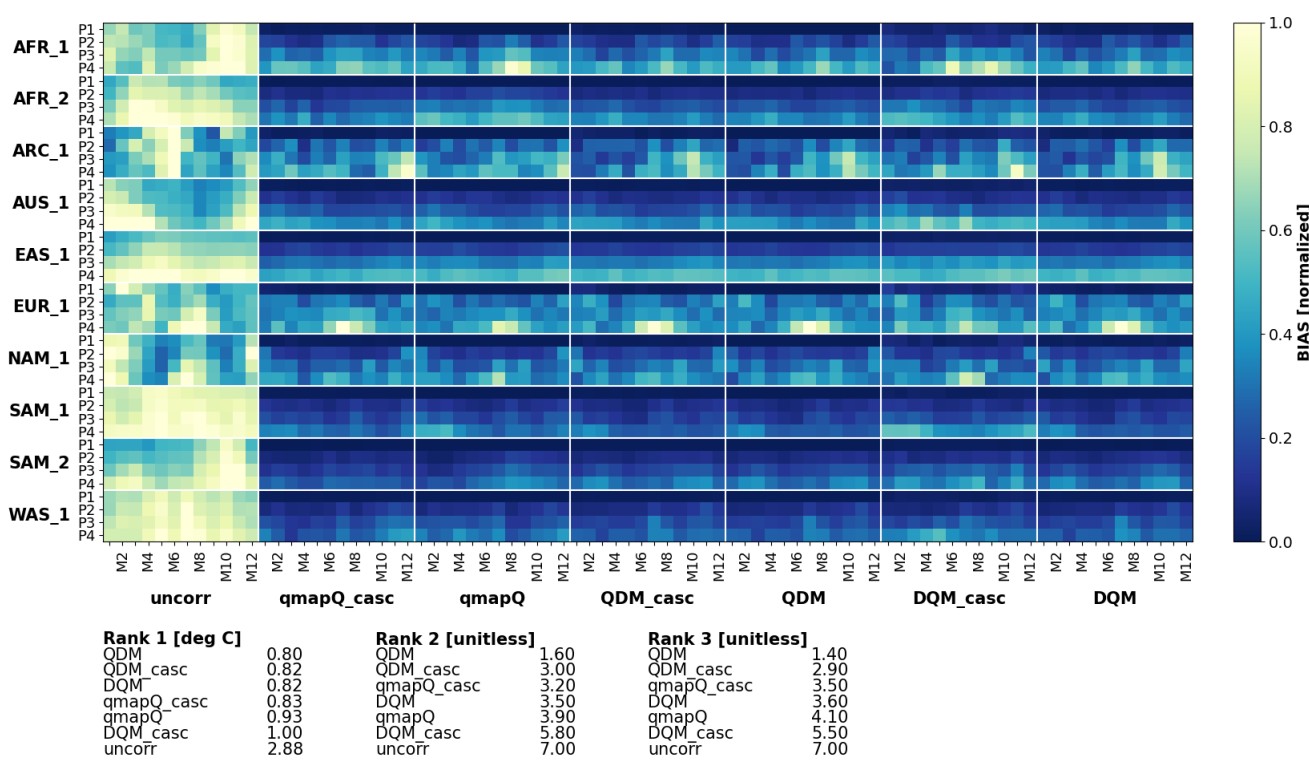

**Figure 5.** Same as Fig. 4, but for mean temperature.

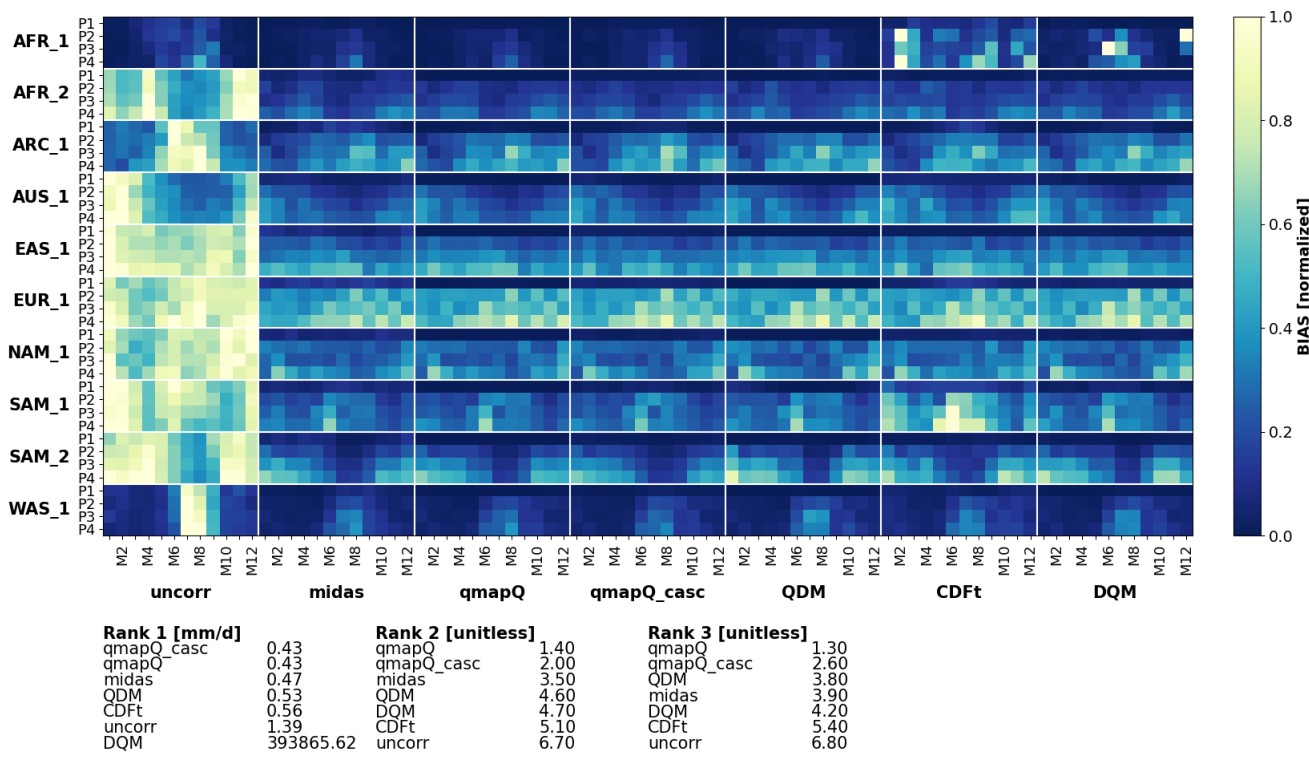

**Figure 6.** Bias of mean precipitation in the original data (uncorr) and after applying the different bias adjustment methods. P1–4 mark the evaluation periods 1971-2000 (also calibration period), 2011-2040, 2041-2070, 2071-2100. The results are presented for each of the domains (vertical) and for each month of the year (M1–12). The results are normalised for each sub-panel, i.e. each domain, such that the bias of the uncorrected data and all methods are in the range 0–1.

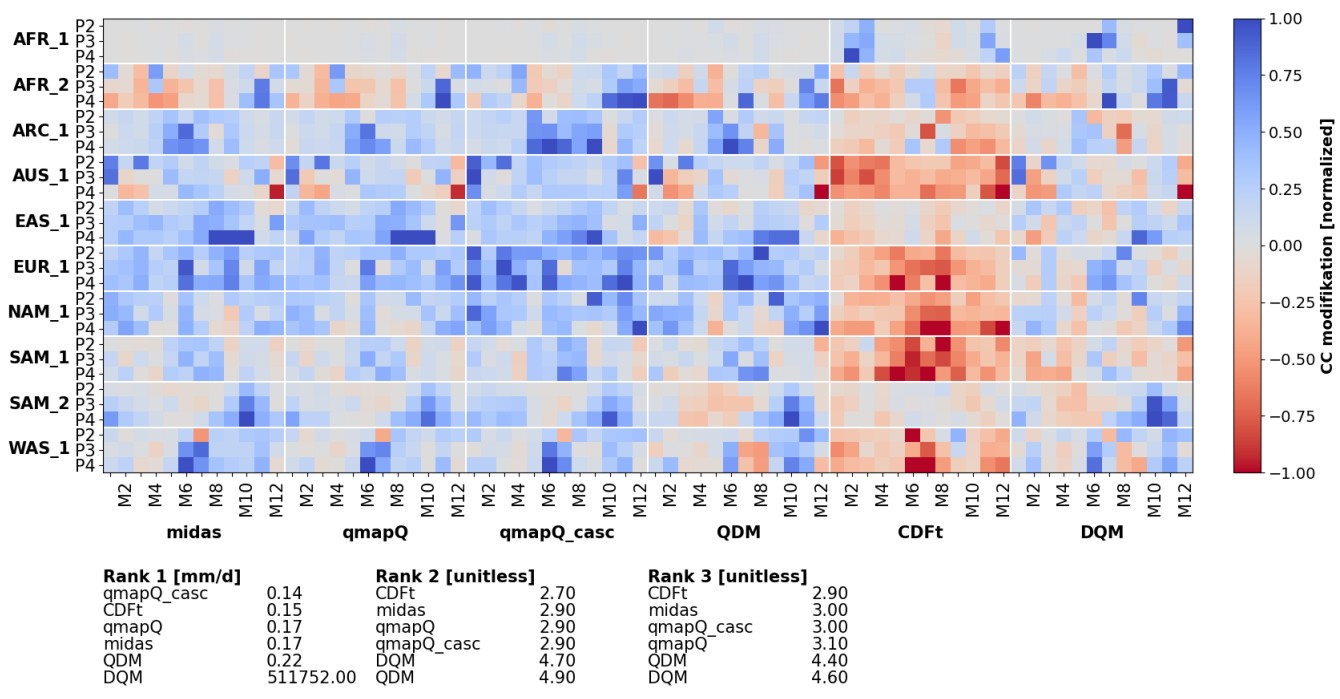

| Rank 1 [mm/d] | | Rank 2 [unitless] | | Rank 3 [unitless] | |
|---|---|---|---|---|---|
| qmapQ_casc | 0.14 | CDFt | 2.70 | CDFt | 2.90 |
| CDFt | 0.15 | midas | 2.90 | midas | 3.00 |
| qmapQ | 0.17 | qmapQ | 2.90 | qmapQ_casc | 3.00 |
| midas | 0.17 | qmapQ_casc | 2.90 | qmapQ | 3.10 |
| QDM | 0.22 | DQM | 4.70 | QDM | 4.40 |
| DQM | 511752.00 | QDM | 4.90 | DQM | 4.60 |

**Figure 7.** Same as Fig. 6, but for the modification of the climate change signal, and excluding the original data.

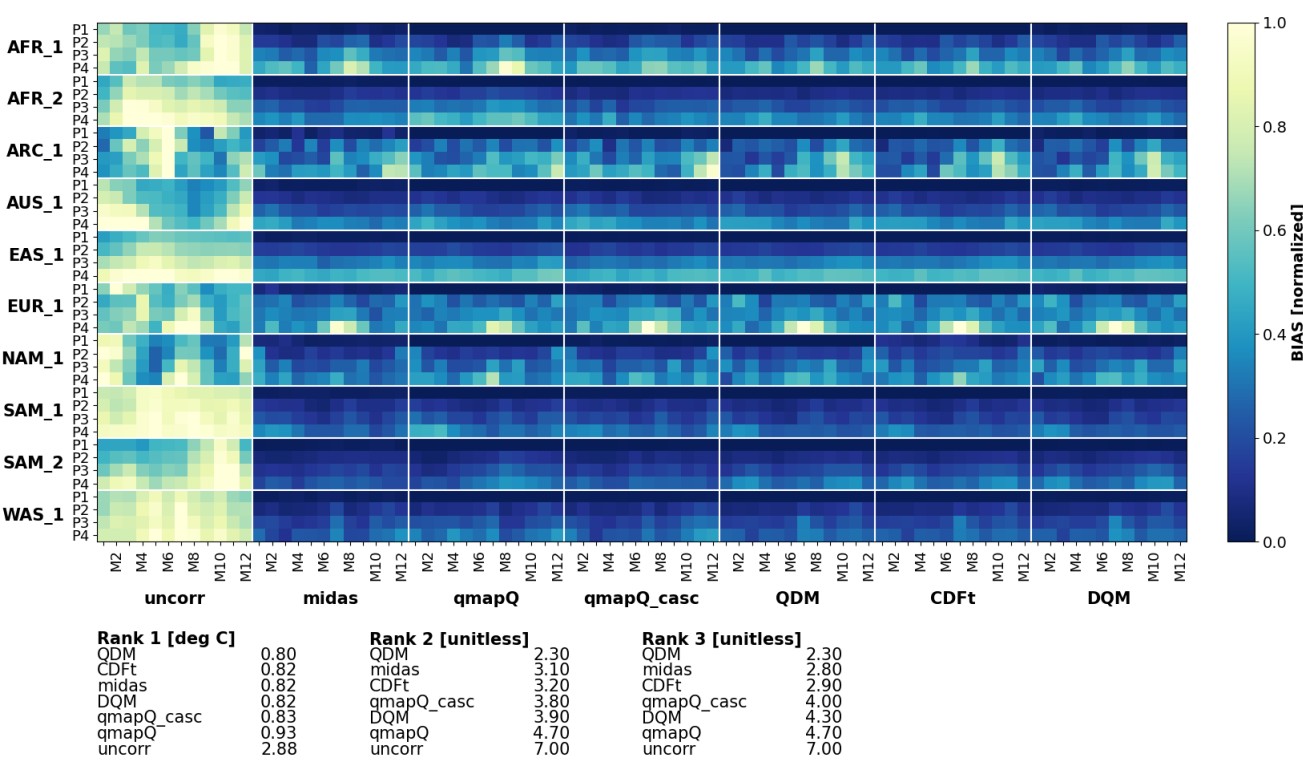

**Figure 8.** Same as Fig. 6, but for mean temperature.

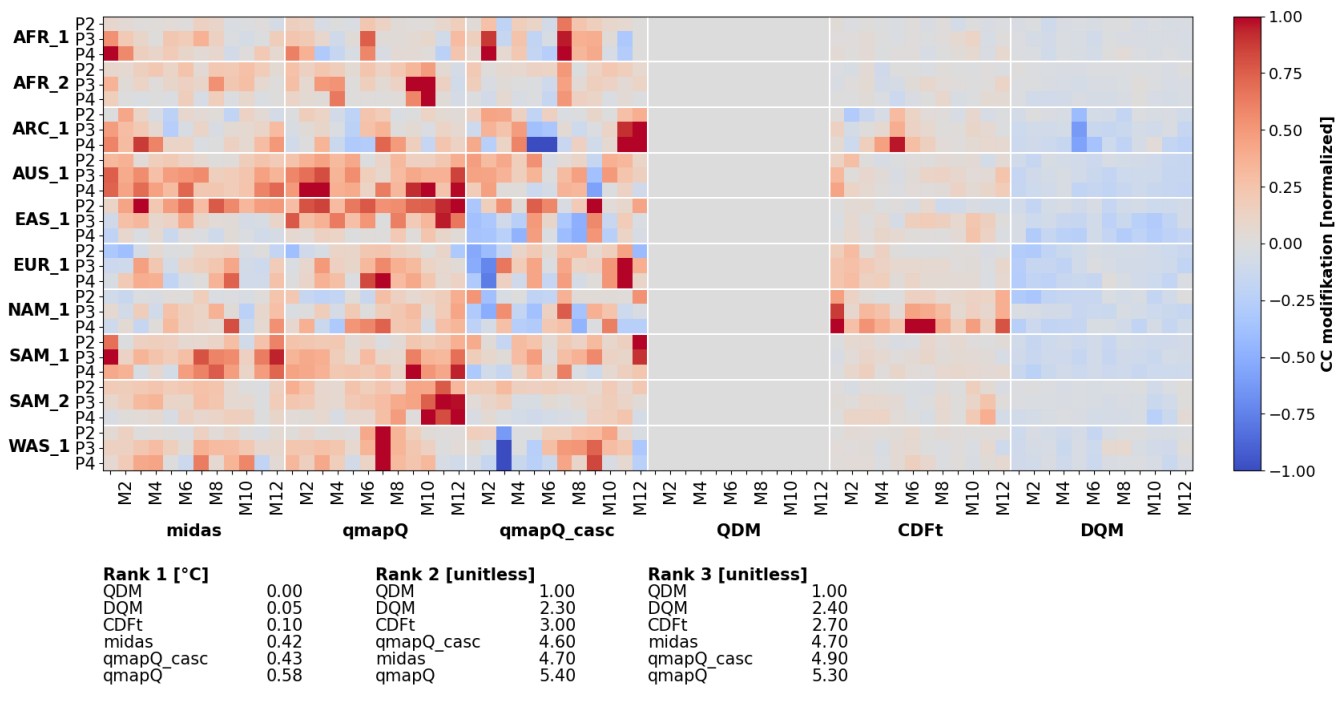

**Figure 9.** Same as Fig. 7, but for mean temperature.