# Peer review of "MIdASv0.2.1—Multi-scale bias Adjustment"

_Geoscientific Model Development, 2022_

## Referee Comment (RC1)

**General comments**

In their contribution, Berg et al. advance cascade bias-adjusting by introducing a practical method for implementing this on both spatial and temporal scales. In my opinion, the importance of this paper is twofold. First, it reinvigorates the cascade bias-adjusting principle, which has, since the papers by Haerter et al. (2010) and Haerter et al. (2011) not received much attention. Second, it does so by introducing applicable code. This allows for discussing some technical details, such as the spline-based fit, which are often overlooked in papers merely introducing new methods. However, when methods are effectively applied, this attention to detail is necessary.

However, I think the focus of the paper is still slightly too narrow and some additional discussions could help other researchers to build on the proposed method. In general, I think two additions could greatly enhance the paper.

First, although Haerter et al. (2010) (and the follow-up paper Haerter et al. (2011), which is strangely enough not mentioned) are the seminal papers regarding cascade bias-adjusting, similar ideas (multiple timescales) are discussed in other papers, such as Johnson and Sharma (2012), Mehrotra and Sharma (2015) and Nguyen et al. (2016). I think it would be fair to acknowledge this and discuss some similarities and differences. In addition, Mehrotra and Sharma (2015) propose a multivariate method for multiple timescales. It would be interesting to have a short discussion on the possibility to allow for multivariate adjustment with MIdAS, as this is a relevant subject nowadays

Second, some limitations are given throughout the text. Some are picked up in the discussion, such as the bounded nature of precipitation, whereas others, such the difficulties with parametric methods, are left out. To further the application and to assess the relevance of MIdAS, it seems relevant to me to have a larger discussion on these limitations.

In addition to these comments, I would like to note that due to time constraints, I could not look at the code and other assets.

**Specific comments**

L. 47: For multivariate adjustment, I think you should also refer to François et al. (2020), as this is one of the few papers so far that provide an overview.

L. 103-125: As far as I know, a spline-based method is not applied anywhere else. While this is implicitly mentioned in the text, it would be good to make this explicit if this is the case. Although this might seem a detail to some, it is indeed a practical and relevant technical improvement for easy bias adjustment.

L. 136: Why are spatial cascades not implemented in MIdAS? Are there any plans to do so in the near of far future?

L. 153: As mentioned in the general comments, it would be interesting to have a more extensive discussion on the benefits and drawbacks of parametric methods and their inclusion in (generally) cascade frameworks and (specifically) MIdAS.

L. 196-198: multiple papers discuss the issues on climate change signal modification. Although an extensive discussion is unnecessary here, it would be great to point the interested reader to some references, such as Maraun (2016), Ivanov et al. (2018) or Casanueva et al. (2018).

L. 203-204: The link between model independence, genealogy and pseudo-reality experiments might not be clear for every reader, given the rather short discussion. It could be relevant to slightly enlarge this.

L. 217-219: These sentences are somewhat confusing, as 'performs slightly worse' contrasts with 'perform worse'. A rephrasing probably allows for more clarity.

L. 221: 'some outlier bias' seems to be contradicted by the large scores for QDM in Figure 5.

L. 226: It could be relevant for some readers to refer to papers discussing the uncertainty in the process chain, such as Bürger et al. (2013), Hingray and Said (2014) or Lafaysse et al. (2014). My knowledge on this topic is far from complete, so other papers might be relevant as well.

L. 278: 'The impact is stronger on other statistics that are not explicitly accounted for.' It is explicitly suggested by Maraun and Widmann (2018) to use statistics that are not accounted for in the evaluation of bias-adjusting methods, which you could refer to.

L. 284 and further: as already mentioned, I miss some information on what could be done to further improve the application of MIdAS, although this could further enhance the relevance of the paper.

L. 290-296: Linked with my comment on L. 136: are there any plans to implement spatial cascaded in the publicly available version of MIdAS? If not in the near future, why?

**Technical comments**

L. 136: 'experiments … has' should be 'experiments have'

L. 147: 'ny physical meaning' should be 'no physical meaning'

L. 174: This line misses a space after '(Maraun, 2012).'

L. 201: 'apart' instead of 'appart

Tables 3 and 4: 'Variable' instead of 'variabel'

Table 4: a bold font is mentioned in the caption, but not applied anywhere in the table.

**References**

Bürger, G., Sobie, S.R., Cannon, A.J., Werner, A.T., Murdock, T.Q., 2013. Downscaling extremes: an intercomparison of multiple methods for future climate. Journal of Climate 26 (10), 3429–3449.

Casanueva, A.; Bedia, J.; Herrera, S.; Fernández, J. & Gutiérrez, J. M., 2018: Direct and component-wise bias correction of multi-variate climate indices: the percentile adjustment function diagnostic tool. Climatic Change, 147, 411-425

François, B., Vrac, M., Cannon, A. J., Robin, Y. Allard, D., 2020. Multivariate bias corrections of climate simulations: Which benefits for which losses? Earth System Dynamics 2020 (2), 1–41.

Haerter, J. O., Berg, P., and Hagemann, S., 2010: Heavy rain intensity distributions on varying timescales and at different temperatures, J. Geophys. Res., 115, D17102, doi:10.1029/2009JD013384.

Haerter, J., Hagemann, S., Moseley, C. Piani, C., 2011. Climate model bias correction and the role of timescales. Hydrology and Earth System Sciences 15, 1065–1073.

Hingray, B. & Saïd, M., 2014: Partitioning internal variability and model uncertainty components in a multimember multimodel ensemble of climate projections. Journal of Climate, 27, 6779-6798

Ivanov, M. A.; Luterbacher, J. & Kotlarski, S., 2018. Climate model biases and modification of the climate change signal by intensity-dependent bias correction. Journal of Climate, 31, 6591-6610

Johnson, F. Sharma, A., 2012. A nesting model for bias correction of variability at multiple time scales in general circulation model precipitation simulations. Water Resources Research 48 (1), W01504.

Lafaysse, M., Hingray, B., Mezghani, A., Gailhard, J., Terray, L., 2014. Internal variability and model uncertainty components in future hydrometeorological projections: the Alpine Durance basin. Water Resour. Res. 50 (4), 3317–3341.

Maraun, D., 2016. Bias correcting climate change simulations-a critical review. Current Climate Change Reports 2 (4), 211–220.

Maraun, D. Widmann, M., 2018. Cross-validation of bias-corrected climate simulations is misleading. Hydrology and Earth System Sciences 22 (9), 4867–4873.

Mehrotra, R. Sharma, A., 2015. Correcting for systematic biases in multiple raw GCM variables across a range of timescales. Journal of Hydrology 520, 214–223.

Nguyen, H., Mehrotra, R. Sharma, A., 2016. Correcting for systematic biases in GCM simulations in the frequency domain. Journal of Hydrology 538, 117–126.

---

## Referee Comment (RC2)

**MIdAS- MultI-scale bias AdjuStment**

**Authors**: Peter Berg, Thomas Bosshard, Wei Yang, and Klaus Zimmermann
**Journal:** *GMD*

**1.    Summery**

In this paper, Berg et al. 2022 suggested a modern code set-up that allows for flexible bias adjustment and compared it to different methods based on quantile mapping. This set-up allows for 1-day-of-year-bias 2-cascacade adjustments to prevent from discontinuity and variance inflation in the data. The paper culminates in discussions about the skill of different methods and future directions for advancing MIdAS code implementation.

**2.    General Comment**

The paper is very well-written. It was quite easy to understand and enjoyable to read the paper. I found the story about King Midas and the effort to relate the story to bias adjustment, quite cool. My only issue was 'the extent of discussion' about some matters. I was expecting a little bit more of explanation (e.g., about CDF-t method, why distribution-based methods are not covered or around L286 to 291). I understand that the authors might deliberately opted out of thorough discussions because of the nature of the paper, but in my opinion, such discussions strengthen this paper and make the whole bias adjustment process clearer.

Given the quality of this paper, I suggest **minor revisions** for this paper.

**3.    Specific Comment**

**L3:** I would remove 'distribution based' from this sentence. There are some advanced multivariate methods that are not distribution based.

**L14-L16:** This whole part is a bit unclear to me.

-Can you please clarify: what do you mean by 'spatial focus is put on preservation of trends'?

-What do you mean by more advanced trend preserving method? Do you consider QDM or CDF-t as the advanced method? To me Midas might be as advanced as QDM (simply because I have worked with QDM but have not implemented multiscale bias adjustment).  Thus, isn't advanced a bit subjective here?

Please also consider naming some of the advanced methods.

**L23:** What are some of the side effect adjustment? Please consider naming some.

**L47**: multi-variate features

**L50:** This sentence is unclear to me. please consider re-writing it. What do you mean by stress test of methods?

**L77:** I would clearly state that why only QM-based methods are selected to be compared to MIdAS.

**L115:** please consider referring to Piani and Haerter (2012).

**L136-138:** Which projects? Why? Please consider clarifying.

**L147**: this sentence needs to be rephrased. Ny probably needs to be changed to no

**L153:** This part needs more clarification. Why distribution-based methods are not favored?

Coming from hydrological community, maybe I am biased but among us distribution-based methods are highly favored. This also comes naturally, as distribution-based smoothing is applied in many hydrological studies to smoothen outliers. In fact, in some studies, at least for temperature, Gaussian distribution seemed to perform reasonably well. Note for example Räty et al. (2018).

**L170:** I would prefer a little bit more explanation of the theory of this method as it is the most intricate one.

Change and to an

**L232:** I don't understand why this part (method intercomparing) is located in result section? Doesn't it fit better in the method section? With e.g., **experiment protocol** subheading?

and why the order of describing variables, is changed (in section 4.1 first temperature is explained while in section 4.2.1 first precipitation is described).

Please consider modifying this section.

**L286-287:** this part seems like a very important part of discussion. However, it is not entirely clear to me what do you mean by different methods for mapping. By 'such methods' which methods are you referring to? Please consider rephrasing.

Sincerely

Faranak Tootoonchi

**References:**

Piani, C. and Haerter, J. O.: Two dimensional bias correction of temperature and precipitation copulas in climate models, Geophys. Res. Lett., 39(20), 1–6, doi:10.1029/2012GL053839, 2012.

Räty, O., Räisänen, J., Bosshard, T. and Donnelly, C.: Intercomparison of Univariate and Joint Bias Correction Methods in Changing Climate From a Hydrological Perspective, Climate, 6(2), 33, doi:10.3390/cli6020033, 2018.

---

## Author Comment (AC1)

**We appreciate very much the comments from all three reviewers, which helps to improve the paper. Below, we respond (in bold font) to all points raised in the review process, including the three full reviews.**

**With best regards,**

**Peter Berg et al.**

RC1: 'Comment on gmd-2022-6', Jorn Van de Velde, 31 Mar 2022

General comments

In their contribution, Berg et al. advance cascade bias-adjusting by introducing a practical method for implementing this on both spatial and temporal scales. In my opinion, the importance of this paper is twofold. First, it reinvigorates the cascade bias-adjusting principle, which has, since the papers by Haerter et al. (2010) and Haerter et al. (2011) not received much attention. Second, it does so by introducing applicable code. This allows for discussing some technical details, such as the spline-based fit, which are often overlooked in papers merely introducing new methods. However, when methods are effectively applied, this attention to detail is necessary.

However, I think the focus of the paper is still slightly too narrow and some additional discussions could help other researchers to build on the proposed method. In general, I think two additions could greatly enhance the paper.

First, although Haerter et al. (2010) (and the follow-up paper Haerter et al. (2011), which is strangely enough not mentioned) are the seminal papers regarding cascade bias-adjusting, similar ideas (multiple timescales) are discussed in other papers, such as Johnson and Sharma (2012), Mehrotra and Sharma (2015) and Nguyen et al. (2016). I think it would be fair to acknowledge this and discuss some similarities and differences. In addition, Mehrotra and Sharma (2015) propose a multivariate method for multiple timescales. It would be interesting to have a short discussion on the possibility to allow for multivariate adjustment with MIdAS, as this is a relevant subject nowadays

**Thanks for the useful references and valid comment to extend the discussion. And great that you spotted that the wrong Haerter et al. paper was referenced! We are at all times referencing the 2011-paper, which is the only one of the two that deals with bias adjustment.**

**We will also mention the three papers that address the different timescales for monthly to annual data in the discussion about how to address the timescales issue.**

**Addressing multivariate methods have been one of the purposes of developing MIdAS, and remains one of the goals. We will add the provided reference, and other reviews on the topic suggested by the other reviewers.**

Second, some limitations are given throughout the text. Some are picked up in the discussion, such as the bounded nature of precipitation, whereas others, such the difficulties

with parametric methods, are left out. To further the application and to assess the relevance of MIdAS, it seems relevant to me to have a larger discussion on these limitations.

**Indeed, it makes sense to discuss these issues in more depth in the discussion section, which we will do in the revision.**

In addition to these comments, I would like to note that due to time constraints, I could not look at the code and other assets.

Specific comments

L. 47: For multivariate adjustment, I think you should also refer to François et al. (2020), as this is one of the few papers so far that provide an overview.

**Yes.**

L. 103-125: As far as I know, a spline-based method is not applied anywhere else. While this is implicitly mentioned in the text, it would be good to make this explicit if this is the case. Although this might seem a detail to some, it is indeed a practical and relevant technical improvement for easy bias adjustment.

**There are examples of spline-based methods, such as the qmap-package in R (the method fitQmapSSPLIN), but we will further emphasize the practical relevance of this approach as suggested in the major comments.**

L. 136: Why are spatial cascades not implemented in MIdAS? Are there any plans to do so in the near of far future?

**Because we haven't yet fully explored the method for this, we have not made a full implementation. The first test cases with this splitting can most easily be done in a pre-processing step as mentioned in the paper. The full implementation will await a more sophisticated approach that can be done on the fly in the code.**

L. 153: As mentioned in the general comments, it would be interesting to have a more extensive discussion on the benefits and drawbacks of parametric methods and their inclusion in (generally) cascade frameworks and (specifically) MIdAS.

**This is an important point, and there are strong practical considerations that are necessary to decide between parametric and non-parametric methods. We mention this, but agree that it would strengthen this point by developing it further in the discussion section.**

L. 196-198: multiple papers discuss the issues on climate change signal modification. Although an extensive discussion is unnecessary here, it would be great to point the interested reader to some references, such as Maraun (2016), Ivanov et al. (2018) or Casanueva et al. (2018).

**Thanks for the suggestion, we will incorporate these references.**

L. 203-204: The link between model independence, genealogy and pseudo-reality experiments might not be clear for every reader, given the rather short discussion. It could be relevant to slightly enlarge this.

**It is indeed a bit short and we will add more information as well as more reference material to direct the interested reader.**

L. 217-219: These sentences are somewhat confusing, as 'performs slightly worse' contrasts with 'perform worse'. A rephrasing probably allows for more clarity.

**We agree.**

L. 221: 'some outlier bias' seems to be contradicted by the large scores for QDM in Figure 5.

**The sentence refers to DQM, not QDM. DQM has outliers which significantly affect the Rank1 score.**

L. 226: It could be relevant for some readers to refer to papers discussing the uncertainty in the process chain, such as Bürger et al. (2013), Hingray and Said (2014) or Lafaysse et al. (2014). My knowledge on this topic is far from complete, so other papers might be relevant as well.

**Our comment was based on our own experience, and is easy to accept as it is clear that introducing outliers or failing to remove bias in some odd occasions introduces uncertainty that might grow further down the model chain. It is not a main focus here, but we will study the references and might add them here if we find that it adds to this point.**

L. 278: 'The impact is stronger on other statistics that are not explicitly accounted for.' It is explicitly suggested by Maraun and Widmann (2018) to use statistics that are not accounted for in the evaluation of bias-adjusting methods, which you could refer to.

**Thank you, we will add this reference as suggested to further emphasize this point.**

L. 284 and further: as already mentioned, I miss some information on what could be done to further improve the application of MIdAS, although this could further enhance the relevance of the paper.

**This is a good point, and we will add a paragraph on this.**

L. 290-296: Linked with my comment on L. 136: are there any plans to implement spatial cascaded in the publicly available version of MIdAS? If not in the near future, why?

**We would like to first develop our ideas on the spatial cascade further, e.g. considering spatial filters. Nevertheless, the addition of spatial cascading is certainly in the future for MIdAS, even though it is not yet on the planned road map. We cannot give any prediction on when such code would be available, and we have not yet opened the repo for community sharing, but hope to do so in the future.**

Technical comments

**Thank you, we will adopt all comments below.**

L. 136: 'experiments … has' should be 'experiments have'

L. 147: 'ny physical meaning' should be 'no physical meaning'

L. 174: This line misses a space after '(Maraun, 2012).'

L. 201: 'apart' instead of 'appart

Tables 3 and 4: 'Variable' instead of 'variabel'

Table 4: a bold font is mentioned in the caption, but not applied anywhere in the table.

**References**

Bürger, G., Sobie, S.R., Cannon, A.J., Werner, A.T., Murdock, T.Q., 2013. Downscaling extremes: an intercomparison of multiple methods for future climate. Journal of Climate 26 (10), 3429–3449.

Casanueva, A.; Bedia, J.; Herrera, S.; Fernández, J. & Gutiérrez, J. M., 2018: Direct and component-wise bias correction of multi-variate climate indices: the percentile adjustment function diagnostic tool. Climatic Change, 147, 411-425

François, B., Vrac, M., Cannon, A. J., Robin, Y. Allard, D., 2020. Multivariate bias corrections of climate simulations: Which benefits for which losses? Earth System Dynamics 2020 (2), 1–41.

Haerter, J. O., Berg, P., and Hagemann, S., 2010: Heavy rain intensity distributions on varying timescales and at different temperatures, J. Geophys. Res., 115, D17102, doi:10.1029/2009JD013384.

Haerter, J., Hagemann, S., Moseley, C. Piani, C., 2011. Climate model bias correction and the role of timescales. Hydrology and Earth System Sciences 15, 1065–1073.

Hingray, B. & Saïd, M., 2014: Partitioning internal variability and model uncertainty components in a multimember multimodel ensemble of climate projections. Journal of Climate, 27, 6779-6798

Ivanov, M. A.; Luterbacher, J. & Kotlarski, S., 2018. Climate model biases and modification of the climate change signal by intensity-dependent bias correction. Journal of Climate, 31, 6591-6610

Johnson, F. Sharma, A., 2012. A nesting model for bias correction of variability at multiple time scales in general circulation model precipitation simulations. Water Resources Research 48 (1), W01504.

Lafaysse, M., Hingray, B., Mezghani, A., Gailhard, J., Terray, L., 2014. Internal variability and model uncertainty components in future hydrometeorological projections: the Alpine Durance basin. Water Resour. Res. 50 (4), 3317–3341.

Maraun, D., 2016. Bias correcting climate change simulations-a critical review. Current Climate Change Reports 2 (4), 211–220.

Maraun, D. Widmann, M., 2018. Cross-validation of bias-corrected climate simulations is misleading. Hydrology and Earth System Sciences 22 (9), 4867–4873.

Mehrotra, R. Sharma, A., 2015. Correcting for systematic biases in multiple raw GCM variables across a range of timescales. Journal of Hydrology 520, 214–223.

Nguyen, H., Mehrotra, R. Sharma, A., 2016. Correcting for systematic biases in GCM simulations in the frequency domain. Journal of Hydrology 538, 117–126.

####################################################################

####################################################################

**RC2**: ['Comment on gmd-2022-6'](), Faranak Tootoonchi, 11 Apr 2022

Attached please find my review.

**Citation**: https://doi.org/10.5194/gmd-2022-6-RC2

MIdAS- MultI-scale bias AdjuStment

Authors: Peter Berg, Thomas Bosshard, Wei Yang, and Klaus Zimmermann

Journal: GMD

1. Summery

In this paper, Berg et al. 2022 suggested a modern code set-up that allows for flexible bias adjustment and compared it to different methods based on quantile mapping. This set-up allows for 1-day-of-year-bias 2-cascacade adjustments to prevent from discontinuity and variance inflation in the data. The paper culminates in discussions about the skill of different methods and future directions for advancing MIdAS code implementation.

2. General Comment

The paper is very well-written. It was quite easy to understand and enjoyable to read the paper. I found the story about King Midas and the effort to relate the story to bias adjustment, quite cool. My only issue was 'the extent of discussion' about some matters. I was expecting a little bit more of explanation (e.g., about CDF-t method, why distribution-based methods are not covered or around L286 to 291). I understand that the authors might deliberately opted out of thorough discussions because of the nature of the paper, but in my opinion, such discussions strengthen this paper and make the whole bias adjustment process clearer.

**Thank you very much, we are glad that you liked the kind Midas story. We have focused mainly on the technical aspects, but because we hear from several reviewers that more context and discussion would strengthen the paper, we will improve this aspect.**

Given the quality of this paper, I suggest minor revisions for this paper.

3. Specific Comment

L3: I would remove 'distribution based' from this sentence. There are some advanced multivariate methods that are not distribution based.

**The intent is a range of methods, where a distribution based multi-variate method is more advanced than a non-distribution based multi-variate method. We will try the sentence on several colleagues to test if it is interpreted as intended.**

L14-L16: This whole part is a bit unclear to me.

-Can you please clarify: what do you mean by 'spatial focus is put on preservation of trends'?

**"Trends" was in this case in the meaning of "changes between two time-slices". We will consider reformulating to stress this point. It is unclear if there is a typo in your comment, but the paper reads "Special" and not "Spatial", which might alleviate the confusion?**

-What do you mean by more advanced trend preserving method? Do you consider QDM or CDF-t as the advanced method? To me Midas might be as advanced as QDM (simply because I have worked with QDM but have not implemented multiscale bias adjustment). Thus, isn't advanced a bit subjective here?

**True, "more advanced" is unclear. We will reformulate along the lines of "…similar to methods that explicitly preserve trends".**

Please also consider naming some of the advanced methods.

**Yes, we will do this if the character limitations in the abstract allow this after all changes are made, otherwise they are mentioned later in the text.**

L23: What are some of the side effect adjustment? Please consider naming some.

**This is a lead-in to the story of King Midas which leads to the discussion on side effects. However, realizing now that this wording is not repeated later, we will reformulate along line 54 to more strongly emphasize this.**

L47: multi-variate features

**Ok.**

L50: This sentence is unclear to me. please consider re-writing it. What do you mean by stress test of methods?

**Ok, we will clarify this. In this context, by stress testing we mean to trial a method in a large range of climates and biases.**

L77: I would clearly state that why only QM-based methods are selected to be compared to

MIdAS.

**We will state that this is because we want to explore the cascade adjustments for all methods. Further, these are all often applied methods and therefore highly relevant for the comparison.**

L115: please consider referring to Piani and Haerter (2012).

**Thanks.**

L136-138: Which projects? Why? Please consider clarifying.

**This experiment was performed in a yet unpublished study. We will extend this to explain more about the context in which this was performed.**

L147: this sentence needs to be rephrased. Ny probably needs to be changed to no

**Ok.**

L153: This part needs more clarification. Why distribution-based methods are not favored? Coming from hydrological community, maybe I am biased but among us distribution-based methods are highly favored. This also comes naturally, as distribution-based smoothing is applied in many hydrological studies to smoothen outliers. In fact, in some studies, at least for temperature, Gaussian distribution seemed to perform reasonably well. Note for example Räty et al. (2018).

**Our point is that it involves considerable effort to determine the appropriate distribution for the anomalies in the cascades, which is why we opt out of using these methods. See also the answer to your above comment for L77. Further, in our experience a certain distribution might work well in one region and season, but might be less well suited (or worse) in other locations or seasons. An example is the use of a normal distribution when there is persistence around zero degrees due to melting and freezing processes, which will give a "spike" in the distribution.**

L170: I would prefer a little bit more explanation of the theory of this method as it is the most intricate one. Change and to an

**It is a challenging method to describe in a short paragraph, but we will attempt at extending this paragraph.**

L232: I don't understand why this part (method intercomparing) is located in result section? Doesn't it fit better in the method section? With e.g., experiment protocol subheading? and why the order of describing variables, is changed (in section 4.1 first temperature is explained while in section 4.2.1 first precipitation is described).

Please consider modifying this section.

**The methods are presented in the method section, and the results from the inter-comparison in this result section. We find this division both logical and of standard approach. Good point about the presentation order of the variables which can be improved.**

L286-287: this part seems like a very important part of discussion. However, it is not entirely clear to me what do you mean by different methods for mapping. By 'such methods' which methods are you referring to? Please consider rephrasing.

**We will explicitly state that we refer to both interpolation methods (optimal interpolation, kriging etc) and other models such as physical connections to orography (e.g. wind-side and altitude effects).**

Sincerely

Faranak Tootoonchi

References:

Piani, C. and Haerter, J. O.: Two dimensional bias correction of temperature and precipitation copulas in climate models, Geophys. Res. Lett., 39(20), 1–6, doi:10.1029/2012GL053839, 2012.

Räty, O., Räisänen, J., Bosshard, T. and Donnelly, C.: Intercomparison of Univariate and Joint Bias Correction Methods in Changing Climate From a Hydrological Perspective, Climate, 6(2), doi:10.3390/cli6020033, 2018.

#######################################################################

#######################################################################

**RC3**: 'Comment on gmd-2022-6', Joel Fiddes, 19 Apr 2022

General Comments

In this manuscript, Berg and coauthors introduce a new Python-based implementation platform for bias adjustment called MIdAS (MultI-scale bias AdjuStment) with one of the more impressive acronym justifications I've seen in the introduction.

The authors state that this is an extendable platform that is aimed at modern computing infrastructure and workflows and thus makes use of modern libraries such as IRIS and DASK.

The paper is generally well written, the experiment setup is clear and well designed and I think the comparison of the various methods (with particular implementations) is a really useful contribution, particularly the effect of the method on the preservation of the climate change signal, which is often an important concern of bias correction studies.

However, I think some important points of discussion are still currently missing and also a stronger positioning of the contribution, which, I think is the fact that this is designed to be a flexible and modern platform for the implementation of various bias correction methods. I feel this needs to be more strongly demonstrated.

Main comments

The authors very much position this contribution as an extendable platform based on modern computing workflows. Therefore I think it would be useful to have more discussion on the technology itself and which bottlenecks are solved by using specific libraries. This is currently reduced to a single sentence on p3 l.82--84. It actually seems there is more discussion on the technology in the abstract than in the text itself. An example of this is why use IRIS instead of Xarray/Dask. I think an entire section would be justified on this in the Methods Section 2 and then perhaps rename Section 2 so that it is a broader section covering both the theory and the technology.

**Thanks, we agree and will extend the description and discussion about the potential and specifics around these technical solutions.**

Connected to (1), I would really like to see a system diagram or similar that makes it clear how the actual implementation is put together and illustrates the "platform" nature of the software with a view to extendability and how that is in the system design.

**That is a good idea and we will explore this.**

Connected to (1) and (2), more justification of the contribution needs to be made, I think. I see the strength of this paper is the stated aim of developing a modern extendable platform, however, I do not currently see this as being well described or demonstrated. I think this needs to include, for example, a discussion on the vision of the next steps and how that is currently part of the system design, extension to other variables, and also performance bottlenecks that have been solved by the implementation with modern libraries. Arguably, when the platform itself is the novelty rather than the methods implemented therein, I might expect to see a community available development page so progress can be followed.

**We will add presentation/discussion about the extendable platform aspects as it is an important aspect for bias adjustment methods when domain sizes, model resolution, and ensemble sizes grow. Regarding the community building, we have yet to make a decision on whether the code development will be opened up to the public, but certainly hope to be able to do so in the future.**

While it is quite normal that just T and P are targeted in such studies it would be nice to see that motivated and should probably be stated that only T and P are evaluated in either 2.4 or 3 so the reader knows what to expect. Is there an ambition to include other variables?

**Good point, we will make this more apparent to the reader. The presented code has only stated support for these two variables, but development is already including other variables such as wind, relative humidity, radiation, cloudiness, etc. We will mention this in an outlook towards the end of the paper.**

Precipitation frequency is mentioned in Section 2 under "SSR" however there doesn't seem to be any evaluation of this. Given wet day frequency can be an important feature of bias I think it would be useful to see this evaluation.

**This was evaluated in an earlier report (only published in Swedish) and we decided not to include that here to keep the paper from growing too large. We will consider if this can be added, or perhaps re-iterated in English in a supplementary.**

For impact assessments, bias correction often has an implicit downscaling step in order to produce time series at a scale that is meaningful for impact models (e.g. Teutschbein, C., and J. Seibert, 2012, Rajczak et al. 2016, Fiddes et al. 2022 - note this is not a suggestion to cite, merely an example of the point). While there is some discussion of this e.g. P.11 l.287-290. I think this deserves more attention given the importance of such techniques to the impact model community, who are important target users of bias adjustment methods.

**Yes, this is often the case. The presented method was not specifically evaluated for this aspect here (although we know from ongoing activities that it behaves as expected and along the lines of other methods in this regard). We will consider mentioning this aspect in the introduction along with references.**

The effect of bias correction on the climate change signal is an important topic but only first mentioned here on P.7 in the evaluation strategy Section 2.4. I think there needs to be some discussion of this in the introduction and discussion with appropriate citations e.g. Themeßl et al. 2012 (among others).

**Thanks, we will mention this aspect also in the introduction.**

Minor comments

**We will address all the minor comments. Thanks.**

Title: I believe a code version number is required by GMD in the title.

Abstract: I think you can remove the last sentence as the code availability is given at the end of the paper. In addition, I don't think Berg 2021 is the citation you mean as that seems to be a data paper in the reference list.

P.3 l.75 "know" to "known"

P.5 l.147 "ny" to "new"

P.5 l.147-149 consider rephrasing as the sentence structure is quite unclear here.

P.6 l.174 Missing space after full stop. "(Maraun, 2012).In"

P.7 l.197 can you give a citation for this important point ("However, detailed studies….")

P.7 l.197 "However, detailed studies…." - this sentence also needs a restructure I think as it is not currently  grammatically clear.

P.7 Section 3 I think it should be stated here what temporal resolution the data is.

Table 3 and 4: typo "Variabel"

Table 3 and 4: what are the units?

P12 l.308 Suggested sentence structure: "Further, MIdAS has the following additional features as compared to currently released bias adjustment software"

Code comments

Data and code are present and well structured and documented in the two cited Zenodo repositories.

While an open repository is not required by GMD, it may be nice to see if there is a community page where users can submit bugs etc and follow development.

There is no environment.yml as stated in the README.rst.

**Thanks for spotting this! We will add it to the repository.**

However, Pip install worked fine to install the code.

References

Fiddes, J., K. Aalstad, and M. Lehning, 2022: TopoCLIM: rapid topography-based downscaling of regional climate model output in complex terrain v1.1. Geosci. Model Dev., 15, 1753–1768.

Rajczak, J., S. Kotlarski, N. Salzmann, and C. Schär, 2016: Robust climate scenarios for sites with sparse observations: a two-step bias correction approach. Int. J. Climatol., 36, 1226–1243.

Teutschbein, C., and J. Seibert, 2012: Bias correction of regional climate model simulations for hydrological climate-change impact studies: Review and evaluation of different methods. J. Hydrol., 456-457, 12–29.

Themeßl, M. J., A. Gobiet, and G. Heinrich, 2012: Empirical-statistical downscaling and error correction of regional climate models and its impact on the climate change signal. Clim. Change, 112, 449–468.

Citation: https://doi.org/10.5194/gmd-2022-6-RC3

---

## Author Response (AR2)

Dear Fabien,

Thanks for your positive response to our paper and revisions.

The following changes have been made, following your comments below:

1. The equation on L141 was changed, thanks for spotting this error.
2. We have thanks to your comment managed to get publication of the repo authorized and we added a sentence to the code availability section: "The MIdAS git repository is open for all to access and use under the GNU LESSER GENERAL PUBLIC LICENSE v3, at https://git.smhi.se/midas/midas. We welcome participation in the further development of MIdAS by requesting developer access through the git repository"
3. Your points about the restricted access to the zenodo links relates to a mistake from my side which was spotted by your co-worker. I made an update to the zenodo and the manuscript a few days prior to your comment. So it is completely open and accessible without a need for a request.

With this, we submit our revised manuscript.

With best regards,

Peter Berg on behalf of all authors.

*Dear authors,*

*thank you for submitting a revised version of your manuscript. The three reviewers were positive about your work and I have therefore evaluated the revision myself. I am happy to accept this publication on GMD, pending the modifications explained below.*

*Best regards,*

*Fabien Maussion*

*L141 - shouldnt the sum be divided by N to represent the average? (overbar)*

*Regarding Code availability*

*You write on L349: "The idea behind the development of MIdAS was to primarily have a good platform to build bias adjustments methods on."*

*This raises the question: by whom? You haven't replied to Reviewer #3's comment on code availability on a development platform, which I agree with. While GMD does not enforce the code to be on gitlab/github, from a reader's and potential user's perspective, it is very unclear what the project governance of MIDAS will look like in the future. Your choice of GPL licensing does not prevent anyone to extend your code and put it themselves freely on the internet, so I would highly recommend to do it yourself before someone else does.*

*Furthermore, I kindly request your code to be freely available on Zenodo (not upon*

*request pending approval based on an email address). This mechanism of approval might give a sense of control but is detrimental to open science for a few reasons:*
*- it is not clear who you will grant access to and who you wont, and why. For example, will friends have access but foes won't?*
*- what happens if you change job or interests? Who will grant access in this case?*
*- requesting access is an obstacle to scientific curiosity and discovery since it might discourage readers who don't want to have their email address to be stored somewhere without a clear privacy policy.*
*- finally, because of your choice of a free license, granting access becomes obsolete the day someone else puts the code online.*

*Please don't hesitate to contact me directly if you have questions or doubts regarding the points above.*